# Patient-Derived In Vitro Models of Ovarian Cancer: Powerful Tools to Explore the Biology of the Disease and Develop Personalized Treatments

**DOI:** 10.3390/cancers15020368

**Published:** 2023-01-05

**Authors:** Chiara Battistini, Ugo Cavallaro

**Affiliations:** Unit of Gynaecological Oncology Research, European Institute of Oncology IRCCS, 20139 Milan, Italy

**Keywords:** ovarian cancer, patient-derived models, cancer stem cells, spheroids, organoids, organotypic cultures, tumor microenvironment, personalized medicine

## Abstract

**Simple Summary:**

Ovarian cancer (OC) is a highly lethal neoplasm with a poor rate of response to current treatment. A prerequisite to designing and validating innovative and more efficacious therapies is a deeper understanding of the biological mechanisms that underlie OC progression and chemoresistance. Such an objective, in turn, requires experimental models that mimic the disease as faithfully as possible. In this context, great help comes from the use of patient-derived material, which is key to establishing clinically relevant models. This review summarizes the different categories of in vitro patient-derived OC models and outlines their ability to represent specific aspects of OC biology and provide new tools for personalizing the treatment of such a devastating disease.

**Abstract:**

Epithelial ovarian cancer (OC) is the most lethal gynecological malignancy worldwide due to a late diagnosis caused by the lack of specific symptoms and rapid dissemination into the peritoneal cavity. The standard of care for OC treatment is surgical cytoreduction followed by platinum-based chemotherapy. While a response to this frontline treatment is common, most patients undergo relapse within 2 years and frequently develop a chemoresistant disease that has become unresponsive to standard treatments. Moreover, also due to the lack of actionable mutations, very few alternative therapeutic strategies have been designed as yet for the treatment of recurrent OC. This dismal clinical perspective raises the need for pre-clinical models that faithfully recapitulate the original disease and therefore offer suitable tools to design novel therapeutic approaches. In this regard, patient-derived models are endowed with high translational relevance, as they can better capture specific aspects of OC such as (i) the high inter- and intra-tumor heterogeneity, (ii) the role of cancer stem cells (a small subset of tumor cells endowed with tumor-initiating ability, which can sustain tumor spreading, recurrence and chemoresistance), and (iii) the involvement of the tumor microenvironment, which interacts with tumor cells and modulates their behavior. This review describes the different in vitro patient-derived models that have been developed in recent years in the field of OC research, focusing on their ability to recapitulate specific features of this disease. We also discuss the possibilities of leveraging such models as personalized platforms to design new therapeutic approaches and guide clinical decisions.

## 1. Introduction

Ovarian cancer (OC) is the seventh most common cancer affecting women and the deadliest gynecological malignancy worldwide [1]. Due to a lack of specific symptoms, it is often diagnosed at an advanced stage, resulting in an unfavorable prognosis and a poor 5-year overall survival [2]. OC is highly heterogenous and comprises different diseases with distinct histological, genomic, and prognostic profiles. The majority of cases have an epithelial origin, although they present various histological subtypes, which have been classified into two main categories integrating histopathological, molecular, and genetic characteristics. Type I tumors comprise low-grade serous (LGSOC), mucinous, endometrioid, clear cell, and transitional carcinomas; these tumors are characterized by slow growth and a better prognosis than type II tumors. This latter group presents more aggressive behavior and comprises high-grade serous ovarian carcinoma (HGSOC), undifferentiated carcinoma, and carcinosarcoma [3].

HGSOC is the most frequent type of OC (70%) with a survival rate of only 30% at 5 years [4]. The standard of care for HGSOC is surgical cytoreduction followed by platinum-based chemotherapy. However, 70% of HGSOC relapse within 2 years [5] and almost all recurrent HGSOC ultimately develop chemoresistance and become unresponsive to standard treatments [6]. 

Very few alternative therapeutic strategies have been designed so far for the treatment of recurrent HGSOC, mainly due to the lack of common mutations that could become actionable targets. In fact, apart from TP53 mutations, which are present in almost 95% of cases, no shared genomic alterations have been identified in HGSOC patients until recently [7]. Only in recent decades has it been discovered that approximately half of the patients with HGSOC have defects in their DNA homologous recombination (HR) machinery. Based on the concept that the inhibition of poly-ADP ribose polymerase (PARP), a mediator of single-stranded DNA break repair, leads to synthetic lethality in tumor cells with the homologous recombination deficiency (HRD) phenotype [8], PARP inhibitors (PARPi) have been introduced in clinical management as maintenance therapy, increasing progression-free survival in advanced OC patients [9].

The development of recurrent chemoresistant disease is not the only reason for the poor clinical outcome of HGSOC patients. Another reason is the transcoelomic dissemination and spreading of the disease in the peritoneal cavity. Indeed, tumor cells shed from the primary tumors and are able to reach the peritoneum and the omentum (which are the main sites of HGSOC metastatization) and all the organs of the abdominal cavity carried by the peritoneal fluid, or by ascites, which is frequently present in advanced-stage OC patients [10]. 

The main players in the processes of metastatization and recurrence are ovarian cancer stem cells (OCSCs, also known as cancer-initiating cells, CICs). By definition, OCSCs have the ability to self-renew and initiate tumorigenesis; moreover, they are intrinsically resistant to cytotoxic treatments, thereby they survive standard chemotherapy, and are responsible for tumor re-growth and relapse [6].

Given this complex scenario, there is an urgent need for preclinical models able to capture and faithfully reproduce all the aspects of EOC and amenable to be used for efficient drug discovery. Since most of the traditional immortalized cancer cell lines have proven unable to recapitulate the behavior of such a heterogenous disease [11], there is a growing interest in the use of patient-derived materials, and various patient-derived models have been developed in recent years. This review will describe the various patient-derived models currently used in OC research (Figure 1), focusing on their potential to address different aspects of the disease, from tumor heterogeneity to the cross-talk between tumor cells and the microenvironment, and their applicability for drug discovery and personalized medicine approaches.

## 2. 2D Primary Cell Cultures: Recapitulating Basic Features of the Disease

In OC, primary cell cultures can be established from malignant ascites or from the mechanical/enzymatic dissociation of solid tumors. Tumor cells can be cultured in 2D adherent conditions as primary short-term cultures or immortalized to generate cell lines with unlimited replicative potential. The immortalization of OC cell lines is often time-consuming and presents a low success rate. Indeed, 10 years of various attempts and optimization were needed by Ince and colleagues to define the proper culture conditions that allowed them to generate (and make available to researchers) a panel of 25 immortalized OC cell lines, derived from primary tumors with different histology [12]. These cell lines mirror the genomic features of the original tumors, and apparently, long-term culture does not affect their phenotype. However, the possibility of a progressive divergence from the tissue of origin due to genetic drifts and clonal selection should always be taken into consideration and carefully ruled out (i.e., by single cell-RNAseq analysis). In fact, the accumulation of such events may render OC cell lines too “distant” from the original tumor to be a reliable model for predicting its clinical behavior, and in particular, its response to pharmacological treatments [13,14]. This issue is even more relevant for a highly heterogeneous disease such as HGSOC. Indeed, a number of studies have shown how cell lines commonly used as HGSOC models fail to recapitulate the major genomic features of the disease, such as TP53 mutations [11,15]. 

On the other hand, the establishment of primary cultures from OC specimens has a high success rate of up to 90% [16,17]. In addition, the short-term nature of primary cultures makes them less prone to accumulating genetic alterations and therefore a more reliable model for the tumor of origin. These cultures, however, share some limitations with immortalized cell lines, such as the loss of the native tissue architecture and the lack of a tumor microenvironment [13]. On the other hand, the presence of a single cell type, namely, tumor cells, renders primary cultures amenable for the application of several omics-based analyses of the epithelial compartment of the tumor, which could help in identifying relevant signatures of this disease [18,19]. For example, Francavilla and colleagues analyzed the phosphoproteomics profile of ex vivo primary OC cells compared to normal tissue and this approach unveiled the presence of a cancer-specific signature (i.e., the phosphorylation of the cyclin-dependent kinase CDK-7 and the activation of its target, RNA polymerase II) that was validated in tumor specimens by immunohistochemistry, demonstrating the ability of primary cells to recapitulate in vitro the activation of signaling pathways relevant for OC proliferation [18].

Primary cell cultures can also be employed for in vitro functional assays of clinical relevance, such as the assessment of HR status by measuring RAD51 foci formation as part of the DNA damage response [17]. Another application of primary cultures is the analysis of the crosstalk between tumor cells and the immune compartment. For example, immune cells can be derived from the same patient, and autologous co-cultures can be established to investigate the efficacy of adoptive immunotherapy in OC. This made it possible to demonstrate the cytotoxic effect of CIK (cytokine-induced killer) cells expanded from human peripheral blood lymphocytes on autologous primary cell cultures [20,21,22]. Other studies have shown the possibility of deriving NK cells from ascites and tumor-infiltrating lymphocytes both from ascites and solid biopsies of OC patients, and, after ex vivo expansion, these immune cell populations were able to recognize autologous tumor cells in vitro, exerting a potent cytotoxic effect [23,24].

## 3. 3D Spheroids: Addressing the Role of Cancer Stem Cells

Once primary OC cell cultures have been established, it is possible to generate 3D spheroid cultures from them, exploiting fundamental properties of OCSCs, namely their ability to resist *anoikis*, self-renew, and proliferate when seeded at a low density in non-adherent conditions with a serum-free medium (thus generating clonal spheroids). Such stringent culture conditions allow for the enrichment of primary tumor cultures for cells endowed with stem-like features. Indeed, it has been demonstrated that cells derived from spheroids express high levels of stem genes and exhibit high tumor-initiation ability [25,26]. In the case of OC, 3D spheroids are particularly attractive because they morphologically resemble aggregates of tumor cells present in ascites and allow researchers to study the stem cell compartment of tumors, which is responsible for dissemination, recurrence, and chemoresistance [6,14].

In this regard, patient-derived 3D spheroids have been shown to be more resistant to platinum than their adherent counterpart [25], and therefore they can be used as platforms for drug screening, aimed at identifying compounds not only targeting bulk tumor cells but also OCSCs. Moreover, profiling these cells at the genomic or transcriptomic level may shed light on important aspects of HGSOC biology and unveil relevant mechanisms involved in HGSOC initiation, progression, and stemness.

For example, Lupia et al. profiled the transcriptome of patient-derived OCSCs and identified CD73 as a marker and a driver of stemness in HGSOC [26]. Interestingly, this finding may be clinically relevant because CD73 is also involved in the production of extracellular adenosine, which exerts immunosuppressive functions [27]. Therefore, targeting CD73 may simultaneously disrupt stemness and immune escape, hopefully exerting an efficient anti-tumor effect.

Two studies have been recently published with a thorough comparison of transcriptional and genomic profiles of matched primary ascites, 2D and 3D spheroids cultures [28,29]. 

The work by Lupia and colleagues [28] showed how the genomic and transcriptomic analysis of a set of samples derived from a single HGSOC patient unveiled signatures associated with different steps of HGSOC progression. In fact, from the ascites of a chemo-naïve HGSOC patient, fresh tumor aggregates were isolated and propagated in vivo as patient-derived xenografts (PDXs), mimicking tumor progression, and in vitro as adherent cells (which underwent spontaneous immortalization with passages, maintaining the phenotype of the original tumor) and clonal 3D spheroids enriched for OCSCs. The transcriptomic analysis of all these patient-matched samples allowed to identify three sets of genes specifically enriched in some “states” of the tumor and not in others: (i) genes associated with proliferation increased in OCSCs and PDXs, suggesting the expansion of a pool of cells endowed with tumor-initiating ability, (ii) genes associated with inflammation and the host response, which were expressed in the original ascites and in PDXs, and (iii) genes associated with stemness and the epithelial–mesenchymal transition (EMT), which were expressed in the original ascites and further enriched in OCSCs. Importantly, an analysis conducted on the TCGA-HGSOC dataset showed that the simultaneous expression of genes belonging to these three clusters correlated with poor prognosis, demonstrating that the analysis of a set of samples derived from a single patient was able to capture clinically relevant molecular traits of HGSOC. Moreover, this work shed light on the plasticity of cells derived from a single malignant ascites, which could be propagated in different in vitro and in vivo models, recapitulating several aspects of the original disease [28].

A similar approach was followed by Velletri and colleagues [29], who analyzed at the single-cell level the transcriptome of freshly ascites-derived cells, 2D adherent cultures, and 3D monoclonal spheroids (single cell-derived metastatic ovarian cancer spheroids, or sMOCS) from five HGSOC patients. Compared to 2D cultures, 3D sMOCS better retained—and enriched with subsequent passages—the stem-related features of fresh ascites. Moreover, sMOCS maintained the expression of patient-specific genes and displayed pronounced differences in sensitivity to carboplatin treatment [29], highlighting the superiority of 3D spheroids, compared to 2D adherent cultures, to capture the heterogeneity of HGSOC and the plasticity of OCSCs.

## 4. Patient-Derived Organoids: Capturing Tumor Heterogeneity for Drug Discovery and Study Therapy Resistance

In general, the superiority of 3D vs. 2D cultures in reproducing the physiological cell–cell and cell–matrix interactions is now well established, as is the improved ability of 3D models to mirror the response to therapies of real tumors [30]. In this regard, another 3D culture system that has attracted growing interest in recent years is the organoid model. Organoids were first described in 2009 by Hans Clevers’ group as 3D structures that originated in LGR5+ murine intestinal stem cells, which were able to proliferate indefinitely in vitro and re-create a mini-organ composed of epithelial cells [31]. This concept has also been applied to tumors and cancer stem cells, and patient-derived organoids (PDOs) have then been established for several tumor types, including OC. After the digestion of the tumor, cancer cells are embedded in a mixture of extracellular matrix (ECM) components (such as Matrigel) and cultured in the presence of various growth factors, hormones, and other compounds [14,32]. Notably, both the type of matrix and the precise mixture of supplements added to the growth medium seem to be key factors in determining the success rate in the establishment and the long-term culture of PDOs (as will be described in this paragraph). It should be stressed that the supplements used for PDO culture may differ from the signals present in the TME of the original patients and can potentially influence the ability of PDOs to mimic the tumor of origin, including its response to drugs. While this issue remains to be investigated, such supplements are needed to support the propagation and differentiation potential of cancer stem cells in vitro, which are a prerequisite for PDO generation. On the one hand, the low frequency of cancer stem cells in the tissue of origin may be one of the reasons for the low success rate in PDO establishment, and their intrinsic heterogeneity may explain the discrepancy in the composition of culture media among different cohorts (as addressed in this paragraph for Wnt ligands). On the other hand, this fact renders PDOs an ideal model in which to study minimal residual disease and recurrence, which are responsible for the dismal prognosis of OC patients.

Moreover, given the high heterogeneity of OC and the lack of innovative therapeutic strategies, PDOs have attracted huge interest as platforms to perform drug screenings in a personalized way, and several recent studies have made efforts in the optimization of culture conditions and methods for viability assessment.

For example, Phan and colleagues described a method to generate PDOs by seeding a mix of tumor cells and Matrigel in a ring shape around the rim of the wells of 96-well plates. This geometry allowed one to easily change medium, add drug treatments and measure their effects on organoids number, morphology and viability, rendering this platform amenable for high-throughput drug screenings [33].

Hill and colleagues established a platform of short-term patient-derived HGSOC organoids to evaluate the correlation between defects in the DNA repair machinery and sensitivity to different drug treatments. They generated 34 organoid lines from 22 patients with HGSOC (and 1 with LGSOC), from both primary and metastatic lesions. Tumor tissue was digested into small multicellular aggregates that were embedded in Matrigel and grown in media supplemented with various growth factors (including R-spondin 1, suggesting that these PDOs required active Wnt signaling). All assays were conducted on short-term organoid cultures (i.e., 7 to 10 days, 1 or 2 passages). In this time frame, organoids maintained the histological organization, as well as most of the somatic mutations, copy number alterations, and other genomic features of the original tumors. The authors then conducted a thorough analysis of DNA repair defects in their organoid platform, using different approaches. In particular, they tried to correlate the results from genomic analyses (i.e., presence of BRCA1 or 2 mutations or of an HRD mutational signature) with functional properties (i.e., positivity for RAD51 foci post-irradiation as a proxy of HR proficiency, stability of replication forks determined by fiber assays) and sensitivity to a panel of compounds, comprising (i) drugs targeting defects in DNA repair machinery, such as the PARP inhibitor olaparib; (ii) drugs inducing replication fork stalling such as carboplatin, prexasertib (CHK1 inhibitor) and the ATR inhibitor VE-822; (iii) drugs inducing replication stress, such as gemcitabine. Indeed, the authors showed that functional assays were more effective than genomic analysis in predicting the response to different drugs. In fact, irrespective of BRCA1 or 2 mutations, organoids unable to form RAD51 foci were sensitive to PARP inhibition, while organoids positive for RAD51 foci were resistant to olaparib treatment but sensitive to carboplatin, prexasertib, VE-822, and gemcitabine. Notably, the effect of these drugs on the organoids matched the response of the patient of origin. Moreover, the authors demonstrated that sensitivity to carboplatin, prexasertib, VE-822, and gemcitabine was associated with replication fork instability, while carboplatin resistance correlated with fork stability [34]. Overall, this work showed how short-term PDOs may provide an efficient tool for predicting patient response to different therapeutic regimens, and these compelling findings have fueled efforts towards the generation of long-term organoid cultures.

The first report of this achievement came from Kopper et al., who established 56 organoid lines from 32 patients with different histological types of OC, both pre- and post-chemotherapy treatment. Original tumors were dissociated, and tumor cells were embedded in BME (basement membrane extract) instead of Matrigel. Interestingly, some of the organoids required the presence of Wnt in the culture media, while others did not. The authors showed that organoids maintained histological organization and biomarkers expression of the original tumors, as well as genomic features, even after several passages, and they also recapitulated the general genomic landscape of OC in terms of Copy Number Variations (CNV) and common mutations. The organoid platform was then used to assess the sensitivity to a wide panel of drugs, ranging from carboplatin and paclitaxel (commonly used in the clinic) to gemcitabine, inhibitors of the PI3K/AKT/mTOR pathway, PARP inhibitors, and Wee1 inhibitors. Similar to what happens in patients, most HGSOC organoids were sensitive to carboplatin, while organoids from other subtypes were less sensitive. Interestingly, for one patient, the authors derived organoids from the primary chemosensitive tumor and from the recurrent chemoresistant lesion, and the latter showed decreased sensitivity to carboplatin in vitro, thus mirroring the behavior of the original disease [35].

Two more studies have been recently published describing other PDO platforms, which differed from the work by Kopper et al. due to the composition of the culture medium, the protocol for Matrigel embedding (in the Matrigel bilayer organoid culture, or MBOC, described by Maru et al., tumor cells were incubated overnight on solidified Matrigel, and then overlaid with a second layer of Matrigel, after the removal of unattached or dead cells), and the overall efficiency in the derivation of stable cultures. Importantly, all these PDOs recapitulated both histological and genetic characteristics of the original tumors, did not accumulate significant mutations during long-term culture, and were also able to capture intra-tumor heterogeneity [36,37].

The organoid platforms described so far were generated from heterogenous clinical specimens, comprising primary tumors, metastatic lesions from different anatomical districts, and tumors of different types and stages, either chemo-naïve or treated with chemotherapy. Instead, Hoffmann and colleagues generated 15 stable organoid lines from 13 advanced HGSOC patients prior to chemotherapeutic treatments, using samples derived only from peritoneal and omental primary lesions. In this work, they expanded the findings of Kopper et al. that some organoids did not grow in the presence of activated Wnt signaling and clearly demonstrated that the activation of the Wnt pathway in HGSOC organoids strongly decreased the growth ability and percentage of stem cells, down-regulated the expression of stemness genes, and up-regulated genes linked to differentiation, shedding light on pivotal mechanisms required for stem cell maintenance in OC [38].

In some of the above-mentioned studies, the responses of organoids to drugs was compared with that of the original patients, but only for a small number of cases. De Witte et al. investigated the correlation between in vitro and clinical sensitivity in a more systematic manner. Indeed, they selected seven PDOs derived from interval debulking surgery of HGSOC patients, challenged them with carboplatin and paclitaxel and showed a significant correlation between the sensitivity of PDOs and clinical response of matched patients (assessed at the histopathological, biochemical and radiological levels) [39]. Similarly, Gorski and colleagues analyzed the responses of six PDOs to carboplatin and showed a significant correlation with the PFS of the original patients [40].

De Witte and colleagues also showed that it was possible to assess the response of PDOs to a broader spectrum of drugs (from 3 to 17 per PDO) in a maximum timeframe of 3 weeks, which could be compatible with clinical practice (particularly with the diagnosis–treatment interval) [39]. Overall, these studies highlight the ability of PDOs to predict patients’ responses to therapies, paving the way to their use both as platforms for the development of novel therapeutic strategies and as personalized models in which available drug treatments could be tested to guide clinical decisions.

The predictive value of PDOs has been emphasized in many of the above-mentioned studies, and they are commonly perceived as a better system than classical 2D cultures in this regard. However, to our knowledge, a systematic comparison between patient-matched organoids and 2D cells for their ability to mimic the treatment response of the original tumor has not been performed. Thus, the jury is still out for a formal confirmation that PDOs are preferable to conventional cell cultures when attempting to anticipate a patient’s response to therapies.

## 5. Organotypic Omental 3D Models: Mimicking the Cross-Talk between a Tumor and Its Microenvironment

All the patient-derived models described so far—primary 2D adherent cultures, 3D floating spheroids, and 3D organoids—share a major drawback, namely, the absence of the tumor microenvironment (TME), which actually plays a major role in OC progression. Indeed, it is now clear that continuous bi-directional communication takes place between OC cells and their microenvironment: the tumor recruits stromal cells and shapes the TME, which, in turn, sustains the growth, spread, and stemness of cancer cells [41]. This cross-talk may also have major effects on the sensitivity of OC cells to drug treatments. In fact, recent studies have demonstrated that components of the TME, such as the ECM, can profoundly affect OC cell response to chemotherapy [42,43]. Therefore, it is fundamental to generate preclinical in vitro models that incorporate the TME to better mimic the biology of OC and more faithfully predict its response to therapies.

To this end, in vitro 3D organotypic models of ovarian TME have evolved through the years to include almost all the stromal components of the major metastatic site for OC, the omentum. The latter is a large fold of the peritoneum covering the intestine, composed of a monolayer of mesothelial cells covering a sub-mesothelial region that contains an extracellular matrix, blood and lymph vessels, fibroblasts, adypocites, and immune cells [44]. The use of omental organotypic cultures was pioneered by Ernst Lengyel’s group, which established the protocol for isolating primary mesothelial cells and fibroblasts from the omentum and co-culturing them (using a ratio of fibroblasts to mesothelial cells calculated from the analysis of sections of normal omental biopsies) in the presence of ECM components (i.e., Collagen type I), re-creating in vitro the microenvironment encountered by OC cells in their metastatic route [45]. Indeed, with regard to culturing OC cells on these organotypic models, Kenny and colleagues successfully recapitulated the early steps of OC dissemination to the omentum and unveiled relevant molecular mechanisms for the cross-talk between OC and the TME. In fact, they have shown that OC cells secrete TGFβ1, which induces omental mesothelial cells to produce fibronectin [46]. Contact with the mesothelium stimulates OC cells to produce MMP-2, which cleaves fibronectin and vitronectin into small fragments, more efficiently bound by α5β1 and αvβ3 integrins expressed by tumor cells. This ultimately favors tumor cell adhesion to the mesothelium and enhances the invasion of the omentum [47]. 

An added value of these organotypic models is that they can be easily adapted for quantitative high-throughput pharmacological screenings, aimed at identifying compounds that specifically target the ability of OC cells to adhere, invade, or proliferate on the omental TME, without affecting the stromal component, hopefully leading to a durable clinical response [48,49,50].

The omental organotypic models described so far presented two major limitations: first, most of the studies performed previously have exploited OC cell lines for the tumor compartment, thus decreasing the clinical and translational relevance of their findings; second, not all the components of the omental TME (such as endothelial cells, immune cells, and adipocytes) were included in the co-cultures, precluding the possibility to fully address tumor–host interactions that could be relevant for tumor progression. Two recently published studies from Frances Balkwill’s group overcame—at least in part—these issues and described a modular model of omental metastasis in which different cell types can be added, enabling them to investigate their specific functional contribution [51,52]. Adipocytes are major components of the omentum, with a well-established role in sustaining the growth of metastasis [53]; therefore, adipocytes were the first component to be employed in these modular models. In particular, the investigators used adipocyte gels formed by omental adipocytes grown in collagen type I gels. Tri-cultures can then be established by embedding co-cultures of early-passage HGSOC cells and primary omental fibroblasts in the adipocyte gels. These tri-cultures replicate key features of HGSOC patient biopsies in vitro, particularly mimicking ECM remodeling and changes in the expression pattern of ECM components [51]. Building up again on adipocyte gels, Malacrida and colleagues created a tetra-culture model of omental metastasis through the addition of primary fibroblasts first, then primary mesothelial cells, and lastly, early-passage HGSOC cells [52]. Furthermore, they generated penta-cultures, also adding platelets to their models, and showed that platelets promote the production of an ECM associated with poor prognosis and enhanced invasiveness in HGSOC [52]. These elegant multicellular models of omental metastasis have the advantage of including more cell types, thereby faithfully mimicking the interactions between various stromal components and tumor cells, but maintaining a 96-well format that could still allow their use in medium-scale pharmacological screening. Moreover, as stated by the authors, the modular composition allows more cell types to be added in the future, for example, immune cells [51].

## 6. Future Perspectives: Novel Technologies in Patient-Derived Models

The organotypic models that we have described in the previous paragraph added various cellular populations from the omental microenvironment to the cultures of cancer cells, allowing the study of their reciprocal interactions and to better mimic the organization and the behavior of real tumors. However, not all the stromal components are represented in these models; moreover, the tumor microenvironment is not only composed of diverse cell types but it also entails biomechanical properties that should be considered to generate even more reliable in vitro models. In this context, the stiffness of ECM and the shear stress generated by the presence of ascites have been implicated in various events related to OC progression. In this last paragraph, we will focus on new technologies that could enable the introduction of these variables in patient-derived models, in order to increase their ability to recapitulate all aspects of original tumors.

Intriguingly, bioengineered devices can also be exploited to address the question of HGSOC origin. Indeed, Kreeger’s group recently developed a microfluidic device capable of reconstructing the microenvironment (in terms of ECM composition and curvature) of cortical inclusion cysts, which is thought to create a favorable niche for the development of HGSOC [54,55]. Although conducted with murine cells, these studies highlight the intriguing possibility to re-create the precise conditions in which HGSOC originate in vitro, paving the way to future innovative therapeutic approaches, but also putatively to the identification of early diagnostic markers.

During metastasis development, the omentum is dramatically remodeled, both in its cellular components (e.g., adypocites are progressively replaced by fibroblasts and immune infiltration) and in the ECM, leading to an increase in tissue stiffness with disease progression [56]. Moreover, chemotherapy induces ECM remodeling and increases matrix stiffness, which is also able to protect tumor cells from cytotoxic drugs [43]. Given the clear relevance of ECM in various key aspects of OC progression, many studies have employed artificial matrices in which it was possible to manipulate matrix stiffness, such as different types of functionalized hydrogels (reviewed in [57,58]. However, most of these studies relied on OC cell lines, thereby decreasing their impact, and also generated conflicting results about the effect of stiffness on proliferation and spheroid formation. These considerations raise the need to combine such technologies with the use of patient-derived samples, in order to better dissect the functional role of ECM in OC progression and perform drug screenings in clinically relevant settings.

The presence of ascitic fluid, which often accumulates in the peritoneal cavity of advanced-stage OC patients, exposes tumor cells to several mechanical stimuli, such as shear stress, the functional impact of which can be modeled by means of microfluidic devices. Rizvi and colleagues designed a simple microfluidic chip in which cancer cells, carried by the flow, were allowed to adhere to a Matrigel bed and spontaneously formed spheroids. They then demonstrated that the presence of continuous flow for 7 days induced EMT and the acquisition of an aggressive phenotype in tumor spheroids [59]. A similar device was used by Ip and colleagues, although in this case, the channels were coated with poly(2-hydroxyethyl methacrylate) in order to prevent cell adhesion. In this way, it was possible to analyze the behavior of floating tumor spheroids in the presence of shear stress, mimicking cancer cell aggregates in the ascites. Ip and colleagues demonstrated that shear stress increased stemness and chemoresistance of OC spheroids through the activation of the PI3K/Akt pathway [60]. Li and colleagues modified this kind of device by incorporating a cell population of the microenvironment in the model, namely, primary mesothelial cells, which can be cultured as a monolayer in the microfluidic channels, allowing the study of the adhesion of cancer cells to the mesothelium in the presence of shear stress, thus mimicking the interactions between OC cells and mesothelial cells during peritoneal dissemination [61,62].

Microfluidic devices can be further optimized by adding different compartments and re-creating the tissue–tissue interface, which is the basic principle of organ-on-chip technology [63]. The “OvCa-Chip” recently described by Saha et al. consisted of two chambers, one with tumor cells and the other covered by endothelial cells forming a vascular lumen, which was perfused with platelets. The authors showed that the presence of a tumor led to disruption of the vascular barrier, which, in turn, favored platelet extravasation [64]. In a subsequent study, the authors modified the “OvCa-Chip” by adding two chambers containing a Collagen Type I hydrogel (thus mimicking ECM) adjacent to the one with tumor cells. In this “OTME-Chip” (ovarian tumor microenvironment organ-on-chip), they demonstrated that extravasated platelets were able to interact with tumor cells, promoting their invasion towards the ECM chambers and also sustaining proliferation and chemoresistance [65].

Notably, while the studies on microfluidic devices described so far relied on OC cell lines for the tumor compartment, “OvCa-Chips” were also used with primary patient-derived tumor cells [64]. This highlighted the possibility to generate progressively more complete models, integrating different cell types and biomechanical stimuli, and finally creating a platform of patient-derived chips, to be exploited for a personalized approach to OC treatment.

## 7. Conclusions

We have provided an overview of the in vitro models that have been generated so far, starting from patient-derived materials (Figure 1). For each model, we indicated which specific aspect of OC it was better suited to recapitulate, and which clinically relevant information could be derived from specific functional assays. 

Tumor fragments can also be engrafted in immunodeficient mice and propagated in vivo generating platforms of PDXs, which maintain the histological characteristics of the original tumors (their detailed analysis is beyond the scope of this review; for a recent publication, see [14]). Although these in vivo models have major advantages for pharmacological studies, allowing one to assess pharmacokinetic and pharmacodynamic properties and toxicity profiles of the drugs, they present severe drawbacks, from the lack of a host immune system to high costs and ethical concerns. These considerations render in vitro patient-derived models preferable and easier to develop the perspective of personalized medicine. Moreover, notwithstanding the lack of patient-derived models successfully incorporating immune cells at present, we strongly believe that some of the models presented in this review may allow the study of the interaction between the tumor and the immune system in the near future. Indeed, given the unsatisfactory efficacy of immunotherapy in OC treatment up to now, efforts should be devoted to exploiting personalized models in order to understand the reasons for this lack of efficacy and identify positive predictive markers that would help patient stratification.

## Figures and Tables

**Figure 1 cancers-15-00368-f001:**
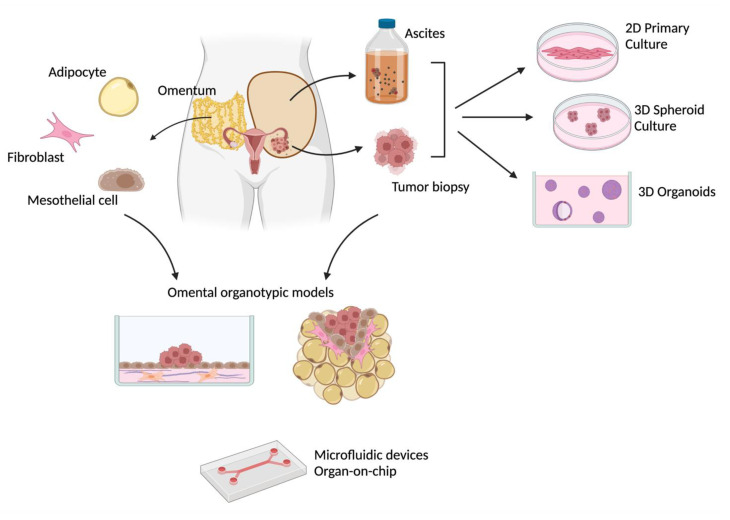
Different patient-derived in vitro models of ovarian cancer. Created with BioRender.com (accessed on 29 November 2022).

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
