# Peer review of "Patient-Derived In Vitro Models of Ovarian Cancer: Powerful Tools to Explore the Biology of the Disease and Develop Personalized Treatments"

_cancers, 2023, doi:10.3390/cancers15020368_

Round 1

Reviewer 1 Report

This is a comprehensive and well written review. One suggestion is to include the microfluidics models by the Kreeger group in UNIV of WISCONSIN. This review will be well cited. 

Author Response

This is a comprehensive and well written review. One suggestion is to include the microfluidics models by the Kreeger group in UNIV of WISCONSIN. This review will be well cited. 

R: We thank the reviewer for her/his positive comments. The models used by the Kreeger group have been added and discussed (lines 395-402).

Reviewer 2 Report

The Review Article titled „ Patient-derived in vitro models of ovarian cancer: powerful tools to explore the biology of the disease and develop personalized treatments” provides a systematic overview of different experimental models derived from primary tumour material and their applications. Considering the significant unmet medical need that this lethal gynecological malignancy still possesses due to severe challenges in therapeutic options such are limited targeted therapies, high recurrence rate, and pervasive chemo resistance this is an important topic and systematic overview is likely to be of value and interest for the researchers in the field. The authors provide insight into diverse 2D primary cultures, 3D spheroid, patient-derived organoids, and microfluidic devices of ovarian cancer highlighting advantages and disadvantages. This systematic approach is very useful as the research field is very diverse and it helps the reader get a better understanding of methodological challenges. The literature is covered with care and comprehensively, though authors should check for the potential inclusion of additional high-impact studies (e.g. Ince et al 2015 Nature Communications).

Also, the manuscript would benefit if more details could be provided on specific methodological readouts that each of the models allows as illustrated by a few examples (types of assays used, quantification strategies, experimental difficulties, etc).

Overall, it is a very well-written review of high quality and it properly highlights the broad potential of the primary models for translational applications in ovarian cancer.

Author Response

The Review Article titled „Patient-derived in vitro models of ovarian cancer: powerful tools to explore the biology of the disease and develop personalized treatments” provides a systematic overview of different experimental models derived from primary tumour material and their applications. Considering the significant unmet medical need that this lethal gynecological malignancy still possesses due to severe challenges in therapeutic options such are limited targeted therapies, high recurrence rate, and pervasive chemo resistance this is an important topic and systematic overview is likely to be of value and interest for the researchers in the field. The authors provide insight into diverse 2D primary cultures, 3D spheroid, patient-derived organoids, and microfluidic devices of ovarian cancer highlighting advantages and disadvantages. This systematic approach is very useful as the research field is very diverse and it helps the reader get a better understanding of methodological challenges. The literature is covered with care and comprehensively, though authors should check for the potential inclusion of additional high-impact studies (e.g. Ince et al 2015 Nature Communications).

R: We thank the Reviewer for her/his positive comments. The study from Ince et al. has been added and discussed (lines 92-102).

Also, the manuscript would benefit if more details could be provided on specific methodological readouts that each of the models allows as illustrated by a few examples (types of assays used, quantification strategies, experimental difficulties, etc).

R: Wherever needed, we have further outlined some methodology-related aspects and issues (e.g., lines 206-217 and 338-339). The readouts are currently detailed, in fact, for all models included in the article.

Overall, it is a very well-written review of high quality and it properly highlights the broad potential of the primary models for translational applications in ovarian cancer.

R: Once again, we wish to thank the Reviewer for her/his positive evaluation.

Reviewer 3 Report

The authors sought to describe existing patient-derived in vitro ovarian cancer models and the features that make them uniquely relevant to studying ovarian cancer. They sought to use cellular characteristics to predict clinical targets for translational research.

2D Primary Cell Cultures

Review of pros and limitations of primary and immortalized cultures from patient-derived samples. No concerns.

3D Spheroids

Reviews evidence of transferrable qualities from patient samples to in vitro 3D growth models. Incorporates some translational aspects by discussing the potential to trial drugs in this model as a method of determining the resistance capacity of cells. No concerns.

Patient-derived organoids

Reviews benefits and translational capabilities of PDOs.

1)    The authors comment on the ability of PDOs to predict response; however, it may be essential to provide distinct evidence of superior prediction ability than other methods or at least some commentary on superiority. Simply predicting response is also achievable with less intensive in vitro models. Moreover, as mentioned, most ovarian cancer patients initially respond to therapy, and the PDO potentially more of an opportunity to research minimal residual disease versus therapy response.

2) It is also essential to address the artificial nature of media and the additives, inhibitors, and growth factors required for PDO development that may not emulate a patient tumor environment. Further, do these additives alter therapy response?

Organotypic omental 3D models

The authors provide a good review of the mechanistic development and potential benefits of this model, but lack data to support its efficacy as a model in translational research.

3) In discussing factors added to this model to develop a more realistic environment, it is important to address the limitation of knowing the ratios of specific cell types that would best recapitulate an in the vivo tumor environment. As such, while adding different cell types is important to recapitulate the heterogeneity of the tumor microenvironment, adding arbitrary cells ratios (e.g., 10 cancer cells with 10 fibroblast cells) will not serve to emulate a tumor and will likely prohibit translation to patient responses.

Future perspective

The authors discuss including stiffness modulation in patient-derived cell lines rather than OC lines and including mechanical stimuli in tumor models. They also discuss tissue interface models as a future direction.

Overall, a well-constructed review of the various in vitro patient-derived models currently in use in ovarian cancer, with a commentary on the benefits of each and future directions to better capture the complexity of ovarian cancer in the laboratory setting. The article does not present any original data, though I think the summarized information is valuable to the scientific community and would recommend publication.

4) As noted by the authors as a limitation to PDX models and knowing that the immune response appears to play a significant role in predicting durable patient responses [1], it's important to address the limitation of in vitro models in general, which is the lack of interaction with the immune system. For instance, PDOs will not allow for immune cell recruitment. Overcoming the ethical challenges associated with in vivo experimental models will be limited by the ability to create a physiologic simulated environment purely in vitro. I think it would be valuable to provide some commentary on the claim of “powerful” in vitro models.

1.         Garsed, D.W., et al., The genomic and immune landscape of long-term survivors of high-grade serous ovarian cancer. Nat Genet, 2022. 54(12): p. 1853-1864.

Author Response

The authors sought to describe existing patient-derived in vitro ovarian cancer models and the features that make them uniquely relevant to studying ovarian cancer. They sought to use cellular characteristics to predict clinical targets for translational research.

2D Primary Cell Cultures

Review of pros and limitations of primary and immortalized cultures from patient-derived samples. No concerns.

3D Spheroids

Reviews evidence of transferrable qualities from patient samples to in vitro 3D growth models. Incorporates some translational aspects by discussing the potential to trial drugs in this model as a method of determining the resistance capacity of cells. No concerns.

Patient-derived organoids

Reviews benefits and translational capabilities of PDOs.

  • The authors comment on the ability of PDOs to predict response; however, it may be essential to provide distinct evidence of superior prediction ability than other methods or at least some commentary on superiority. Simply predicting response is also achievable with less intensive in vitro models. Moreover, as mentioned, most ovarian cancer patients initially respond to therapy, and the PDO potentially more of an opportunity to research minimal residual disease versus therapy response.

R: We thank the Reviewer for these thoughtful comments.

We have now included a reference that highlights the superiority of 3D vs 2D models, both in terms of recapitulating the cellular interactions of the original tumors and of mirroring the drug responses (lines 192-194).

We agree with the reviewer that the predictive value of PDOs is a key issue, and a conclusion on their superiority in this regard should come from their systematic comparison with patient-matched 2D models and original tumor. Unfortunately, such a systematic approach incorporating 2D models has not been reported yet, so no elements to claim the superiority of PDOs are available. This issue has now been included in the manuscript (lines 311-317).

The possibility to use PDOs as models for minimal residual disease has now been pointed out (lines 215-217).

  • It is also essential to address the artificial nature of media and the additives, inhibitors, and growth factors required for PDO development that may not emulate a patient tumor environment. Further, do these additives alter therapy response?

R: The question whether the culture supplements affect the therapy response of PDOs is intriguing. Unfortunately, to our knowledge, none of the published studies on OC-derived PDOs have investigated the effect of these additives on response to drugs. The implications related to the use of culture supplements have now been discussed (lines 206-217).

Organotypic omental 3D models

The authors provide a good review of the mechanistic development and potential benefits of this model, but lack data to support its efficacy as a model in translational research.

  • In discussing factors added to this model to develop a more realistic environment, it is important to address the limitation of knowing the ratios of specific cell types that would best recapitulate an in the vivo tumor environment. As such, while adding different cell types is important to recapitulate the heterogeneity of the tumor microenvironment, adding arbitrary cells ratios (e.g., 10 cancer cells with 10 fibroblast cells) will not serve to emulate a tumor and will likely prohibit translation to patient responses.

R: This is indeed a relevant point, and as a matter of fact the ratio of microenvironment cells used in organotypic models was inferred from the analysis of omental tissue. This has now been specified in the manuscript (lines 338-339).

Future perspective

The authors discuss including stiffness modulation in patient-derived cell lines rather than OC lines and including mechanical stimuli in tumor models. They also discuss tissue interface models as a future direction.

Overall, a well-constructed review of the various in vitro patient-derived models currently in use in ovarian cancer, with a commentary on the benefits of each and future directions to better capture the complexity of ovarian cancer in the laboratory setting. The article does not present any original data, though I think the summarized information is valuable to the scientific community and would recommend publication.

R: We thank the Reviewer for her/his positive evaluation. 

  • As noted by the authors as a limitation to PDX models and knowing that the immune response appears to play a significant role in predicting durable patient responses [1], it's important to address the limitation of in vitro models in general, which is the lack of interaction with the immune system. For instance, PDOs will not allow for immune cell recruitment. Overcoming the ethical challenges associated with in vivo experimental models will be limited by the ability to create a physiologic simulated environment purely in vitro. I think it would be valuable to provide some commentary on the claim of “powerful” in vitro models

R: We agree with the reviewer that incorporating the immune system should become a major goal of next-generation in vitro patient-derived models, and have pointed this out in the manuscript (lines 466-472). Nevertheless, each of the models described here remains “powerful” because of its usefulness in representing specific aspects of the original tumor, thus offering the possibility to investigate and harness specific features, as outlined throughout the article.